# Children’s Play and Independent Mobility in 2020: Results from the British Children’s Play Survey

**DOI:** 10.3390/ijerph18084334

**Published:** 2021-04-20

**Authors:** Helen F. Dodd, Lily FitzGibbon, Brooke E. Watson, Rachel J. Nesbit

**Affiliations:** School of Psychology and Clinical Language Sciences, University of Reading, Reading RG6 6ES, UK; l.t.fitzgibbon@reading.ac.uk (L.F.); brooke.watson@pgr.reading.ac.uk (B.E.W.); r.j.nesbit@reading.ac.uk (R.J.N.)

**Keywords:** child, play, outdoor play, risky play, adventurous play, independent mobility, demographics, green space, nature, playgrounds

## Abstract

The British Children’s Play Survey was conducted in April 2020 with a nationally representative sample of 1919 parents/caregivers with a child aged 5–11 years. Respondents completed a range of measures focused on children’s play, independent mobility and adult tolerance of and attitudes towards risk in play. The results show that, averaged across the year, children play for around 3 h per day, with around half of children’s play happening outdoors. Away from home, the most common places for children to play are playgrounds and green spaces. The most adventurous places for play were green spaces and indoor play centres. A significant difference was found between the age that children were reported to be allowed out alone (10.74 years; SD = 2.20 years) and the age that their parents/caregivers reported they had been allowed out alone (8.91 years; SD = 2.31 years). A range of socio-demographic factors were associated with children’s play. There was little evidence that geographical location predicted children’s play, but it was more important for independent mobility. Further, when parents/caregivers had more positive attitudes around children’s risk-taking in play, children spent more time playing and were allowed to be out of the house independently at a younger age.

## 1. Introduction

It is increasingly recognised that play, including imaginary play, creative play, social play and outdoor play, offers a range of benefits for children’s physical and mental health [1]. Outdoor play, and play in nature in particular, is associated with a range of health benefits [2] including increased physical activity [3] and emotional wellbeing [4,5]. Outdoor play also facilitates children’s opportunities to play in an adventurous way, exploring age-appropriate risks and uncertainty, which is theorised to prevent fears [6] and anxiety [7] in children. Despite these apparent benefits, there is growing concern that children’s opportunities for play, particularly outdoor, adventurous play are diminishing. For example, Clements [8] reported that 70% of mothers surveyed in the United States stated that they played outdoors daily when they were children whereas only 31% of their children were reported to do so. Similarly, 60% of mothers reported playing in adventurous ways, such as climbing trees, whereas only 22% of their children played in this way. This aligns with data from the UK indicating that only 10% of children’s play currently happens in natural spaces such as woodlands, countryside and heaths, whereas parents report that they spent approximately 40% of their play time as children in these natural spaces [9]. Children’s independent mobility, defined as the freedom to travel around their own neighbourhood without adult supervision, is also reported to have reduced dramatically over the past 50 years [10]. In this paper the main findings from the British Children’s Play Survey, a national survey conducted in 2020 with a representative sample of 1919 parents and caregivers, are presented. The paper aims to provide a snapshot of where children aged 5 to 11 years play in Great Britain, how much time they spend playing and how adventurously they play. We also explore socio-demographic and geographic correlates for children’s play, including outdoor and adventurous play.

### 1.1. Previous Large Scale Surveys of Children’s Play

To date, the largest surveys of children’s play in the UK/Great Britain have both focused on play in natural spaces only and have recruited non-representative samples. For example, in 2009, England Marketing produced a report for Natural England which focused on children aged 7 to 11 years. They examined children’s ability to play unsupervised in natural spaces. The majority of children reported that they played indoors at home more than in any other place and, as stated above, that only 10% of children’s play happened in natural spaces such as woodlands, countryside and heaths. Parents expressed that they would like to be able to allow their children to play unsupervised in natural spaces but that concerns about road safety and strangers were barriers to this. More recently Hunt, et al. [11] examined UK children’s time spent in natural environments, such as woodlands, country parks, and rivers/lakes. This survey, which focused on children under the age of 16, showed that 88% had visited the natural environment within the previous 12 months. Children’s visits were predicted by the visits of the adults they lived with; 75% of children who visited the natural environment did so with adults from their household.

Further studies have been conducted in Norway [12], the United States [13] and New Zealand [14]. In Norway, the most commonly used outdoor space was the back garden (69%), although 19% of children played in forest spaces on a daily basis [12]. In the United States, the National Kids Survey [13] showed that the majority of children spent at least two hours outside every day (62.5%) and around half of children spent four or more hours outside on a typical weekend day (51.3%). Importantly, 84% of children said the time outdoors was spent playing or hanging out rather than doing formal activities or sports. The New Zealand State of Play survey focused on risky play, independent mobility and parent attitudes to risk taking in play, using a nationally representative sample of parents with children under 18 years [14]. The results indicated that the majority of children engaged in some risky play activities but that it was relatively rare for children to regularly engage in a broad range of risky play activities, with fewer than 50% of parents reporting that their child ‘often’ or ‘always’ engaged in at least one risky play activity. As children got older, their parents were more likely to allow them to participate in risky play. Parents agreed that there were multiple benefits to be gained from exposure to risk and challenge and that health and safety rules were too strict, but they expressed concern about road safety and stranger danger. In relation to independent mobility, the age at which it was most common for parents to report that their child could go out without adult supervision, but with friends, was 13 years and the age at which it was most common for children to be allowed out entirely alone was 15 years.

### 1.2. Factors Associated with Children’s Outdoor Play and Independent Mobility

Factors that are associated with children’s outdoor play and independent mobility have been examined in a number of studies and systematic reviews. Three groups of factors have been explored—parent behaviour and attitudes, geographic factors, and socio-demographic variables. The most consistent findings relate to parental behaviour and attitudes, highlighting the important role of parents in supporting children’s play behaviour. For example, Ferrao and Janssen [15] found that parent encouragement of outdoor play is a strong predictor of children’s time spent playing outdoors. Parent factors were also highlighted by Veitch et al. [16] who found that in Melbourne, Australia, play at the park was predicted by whether families went to the park together and whether parents were satisfied with the quality of parks and playgrounds in their neighbourhood. Outdoor play also appears to be predicted by geographic factors. For example, a recent systematic review highlighted that features of the built environment are associated with children’s outdoor play: children play outdoors more when there is less traffic, increased neighbourhood greenness and when they have access to a yard [17].

Finally, a number of socio-demographic variables have been identified as predicting outdoor play, although there is some inconsistency between studies. A recent study by Parent et al. [18] found that children in Canada with a European ethnic background and whose families had a higher household income were also more likely to play outdoors in their neighbourhood. A systematic review of parental correlates of outdoor play in children aged 12 and under found that mothers’ ethnicity, mothers’ employment status, parents’ education level, as well as the importance parents assign to outdoor play and neighbourhood social cohesion were associated with outdoor play [19]. Importantly, only one of the studies included in the recent systematic review was conducted in the UK, and that focused on time spent outdoors rather than outdoor play specifically [20]. This UK study showed that boys from lower Socio-Economic Status (SES) backgrounds who played on their computer for less than 2 h on a school day had the highest odds of spending more than 1 h outside after school.

In relation to independent mobility specifically, data from the UK’s Millennium Cohort Study collected between 2007 and 2009 showed that socio-demographic factors associated with independent outdoor play in the UK were being older, male, white British, in poverty, and living close to family and family friends [3]. In Canada, children aged 8 to 12 had greater independent mobility if they were in a higher school grade, spoke English or French at home, were part of a family who did not own a car, and owned a mobile phone. The independent mobility of these children was further predicted by parent perceptions of safety and environment [21].

Taken together, previous research clearly demonstrates that a range of geographic and socio-demographic factors are associated with children’s time spent playing outdoors and their independent mobility. Furthermore, parental attitudes are also likely to influence children’s play, in particular their engagement in adventurous play, as well as independent mobility.

### 1.3. Aim and Research Questions

The overarching aim of the British Children’s Play Survey was to provide nationally representative data on the play of primary school-aged children growing up in Great Britain in 2020. In this paper we address the following research questions:Where do children aged 5–11 years living in Britain spend time playing? Does the amount of time that they spend playing in each place vary significantly?How adventurously do children living in Britain play? Does this vary by play location?What age are children living in Britain allowed out in their neighbourhood alone?To what extent are socio-demographic factors, geographic factors and parental attitudes to risk and protection associated with children’s total time spent playing, time spent playing outdoors and time spent playing adventurously?To what extent are socio-demographic factors, geographic factors and parental attitudes to risk and protection associated with children’s independent mobility?

## 2. Materials and Methods

### 2.1. Participants

The Survey respondents were 1919 parents and caregivers (54% female) of children (49% female) aged 5 to 11 years (M = 8.45, SD = 1.99). Table 1 provides full, detailed demographic characteristics of the sample. The majority of respondents were parents and we therefore refer to them as parents in the tables for ease. A power analysis indicated that a minimum sample of 1229 was required for multiple regression with 10 predictors, a small effect size and 95% power. We exceeded this minimum requirement to ensure that adequate power was maintained even when missing data were considered.

Participants were recruited via YouGov, who are a UK public opinion research company. YouGov operate an online panel that includes over one million adult panelists in the UK and the authors commissioned them to collect data from a nationally representative sample. YouGov recruit panelists from a range of sources to support the diversity of the panel. For the present study, they created a nationally representative sample by drawing on a sub-sample of the national panel that is representative of British adults and inviting them to complete the survey. YouGov use Active Sampling, which means that only panelists invited to participate can complete the survey. Panelists completed the survey online and were given YouGov reward points in compensation for their time. Once fieldwork was complete, the final dataset, including all potential participants who did not complete the survey because they did not have a child within the age bracket of interest, was weighted to the national profile of all adults aged 18+, including those without internet access. The data were weighted based on age, gender, social class, region and level of education.

The methods and procedure were approved by the University of Reading School of Psychology and Clinical Language Sciences Ethics Committee (2020-003-HD). The data and analysis script are available via the UK Data Service: http://doi.org/10.5255/UKDA-SN-8793-1.

### 2.2. Measures

The full British Children’s Play Survey is available via the UK Data Service at http://doi.org/10.5255/UKDA-SN-8793-1. The focus in the present paper is on the measures that specifically ask about children’s play, independent mobility and parental attitudes to risk in play. The wider survey also included questions on children’s organised activities, screen time, mental health and holidays along with questions about respondent’s mental health and some questions related to COVID-19. A series of questions were asked about socio-demographic and geographic characteristics within the survey (see Table 1). In addition, YouGov provided data on respondent ethnicity, whether they reported having had a disability or health problem in the previous 12 months, and whether they lived in an urban, rural or town/fringe area, which had already been collected from respondents as part of their panel membership.

#### 2.2.1. Children’s Play Scale (CPS)

The CPS [23] asks parents about their child’s play in seven places. These are: at home or in other people’s homes; outside at home or at other people’s homes (e.g., garden/yard/balcony); at a playground; in trees/forests/woodland/grassy spaces (not including the garden at home or other people’s homes); in the street or public places close to home; outdoors near water (e.g., at the beach, in the sea, near a river, lake or cliffs); indoor play centres and pools (e.g., soft play, trampoline parks, swimming pools, etc.). For ease of presentation, for the rest of the paper, the ‘trees/forests/woodland/grassy spaces’ category will be referred to as ‘green space’, ‘street or public places’ will be referred to as ‘street’ and the final category will be referred to as ‘indoor play centres’.

Respondents were asked to report, firstly, the frequency with which their child plays in each place, second, the length of time their child plays in each place and, finally, how adventurously their child plays in each place. For frequency and length of time, parents were asked to report for Autumn/Winter and Spring/Summer separately. The frequency questions were answered on a seven-point scale (every day, 4 to 6 times a week, 2 to 3 times a week, once a week, 1 to 3 times a month, less than once a month, never). The time questions were answered on a six-point scale (less than half an hour; half an hour to an hour; 1 to 2 h; 2 to 3 h; 3 to 4 h; more than 4 h). Frequencies were converted into approximate number of days within the six-month seasonal window (e.g., 4 to 6 times a week was converted into 5 times a week on average across 6 months of autumn/winter to give 130 days). Similarly, times were converted into approximate times per day that the child played in the place (e.g., 1 to 2 h was converted into 90 min per day of play). Frequencies and times were multiplied to give the approximate number of minutes a child played in a specific place within a 6-month season. These time estimations for each season were then summed to give a total time spent playing per year variable. In addition, we computed a variable for total time spent playing outdoors, which included five of the seven play locations from the survey, with play indoors at home and at indoor play centres excluded.

The level of adventurous play within each place was rated on a five-point Likert scale ranging from 1 (very low levels of adventure) to 5 (maximum levels of adventure). Definitions of each point on the scale were provided. To calculate a total time spent playing adventurously variable, we calculated the total time spent playing using only places where parents rated that their individual child played with at least a mild level of adventure (2 on the Likert scale).

The test re-test and cross-informant reliability of the three variables from the CPS used in this paper have been evaluated in Dodd et al. [23]. For mothers, test-retest reliability was good, ranging from 0.67 to 0.73. For fathers, this was lower, ranging from 0.39 to 0.49. Cross-informant agreement ranged from 0.37 to 0.51, which is relatively consistent with agreement between informants on well-validated parent-report measures regularly used in public health research e.g., [24].

#### 2.2.2. The Risk Engagement and Protection Survey (REPS)

The REPS is a self-report measure that captures parent and caregiver views and attitudes towards protecting children from injury and allowing them to engage in risks. Respondents report the extent to which they agree with 14 statements such as “Benefits of physical activity for my child outweigh the risk of experiencing minor injuries”. As described in Olsen, et al. [25] we calculated two subscale scores: Protection from Physical Injury (PfI) and Engagement with Risk (EwR). Following Jelleyman et al. [14], these scores were calculated using 12 of the 14 items. Both subscales have a minimum score of 6 and a maximum score of 42. Higher scores indicate greater engagement with risk and greater protection from injury respectively. The internal consistency for both scales was good (PfI alpha = 0.87; EwR = 0.75).

#### 2.2.3. The Tolerance of Risk in Play Scale (TRiPS)

The TRiPS [26] is a self-report questionnaire that measures adult tolerance of risk during children’s play. Respondents are required to answer yes or no to each of 32 items that vary in the how ‘easy’ the items are to endorse. Following the same scoring procedure as Jelleyman et al. [14] a no response is scored as 0, whereas a yes response receives a score from 1–12 weighted according to how acceptable is the level of risk that the items refer to. These scores were determined via a Rasch analysis conducted within the original validation study for the TRiPS [26]. For example, ‘Would you allow your child to play chase with other children?’ is a relatively high acceptability item so would receive a score of 2 and ‘Would you let your child use a hammer and nail unsupervised?’ is an example of a relatively low acceptability item, and would receive a score of 8. A total risk tolerance score was calculated by summing the scores for all 32 items. Possible scores range from 0–184, with higher scores indicating greater risk tolerance. For descriptive purposes, in line with previous work [14], parents were categorised into one of four categories based on their score of risk tolerance: risk averse (0–61), somewhat risk averse (62–95), somewhat risk tolerant (96–122) and risk tolerant (123–184). These categories were determined in a previous study using representative data from New Zealand by dividing scores into quartiles [14].

#### 2.2.4. Age Allowed Out Alone

Respondents were asked to report the age at which they were allowed to play out alone in their local neighbourhood and the age at which they did/will allow their child to play out alone in their local neighbourhood.

### 2.3. Procedure

YouGov panelists were sent an email inviting them to take part in a survey being conducted by YouGov. They were asked to follow the link, where they were given access to the survey. Participants were initially asked whether they had a child within the age range of interest. Those who did not were redirected to a different survey being conducted by YouGov that was unrelated to the present study. Those who had at least one child within the age range were then presented with the rest of the survey questions. They were asked to respond with their eldest child within the age range of 5 to 11 years in mind. Data were collected between 4 April 2020 and 15 April 2020, shortly after the UK-wide lockdown began due to COVID-19. Respondents were clearly instructed to answer the questions thinking about normal life before it was affected by the COVID-19 pandemic at the beginning of the survey and throughout. All respondents were given YouGov points for completing the survey which can be exchanged for payment after reaching a specified number of points.

### 2.4. Missing Data

Some demographic data were missing due to participants choosing not to respond to certain items (see Table 1). This missing data means that the total number of participants included in the models with demographic factors as predictors is substantially reduced from the full sample (see observation numbers shown in results tables). The majority of participants excluded from these analyses (73–78%) were excluded due to missing responses on either ethnicity, respondent health problem/disability or child disability (note that ‘don’t know’ and ‘prefer not to say’ responses were treated as missing). Given that these missing data are unlikely to be missing at random, we decided to run all analyses involving these variables again, excluding ethnicity, respondent health problem/disability and child disability, respectively, to see if results were consistent without the data loss associated with each variable. Removing the predictors from the relevant models provided an overall pattern of results that was very consistent with the results for the reduced sample, which means that we can be confident in the presented results despite the missing data (see analysis script for further details).

For the CPS, there were no missing data for variables capturing time spent playing across place and therefore no missing data for total time spent playing per year or for total time spent playing outside per year. For the questionnaire items on level of adventurous play, participants were only asked to rate how adventurously their child played for those places where they had previously stated that their children played. Any items they did not see were coded as missing. Participants were also able to select ‘don’t know’ for the adventurous play ratings. A ‘don’t know’ response was also coded as missing. This means that for individual places there are missing data for level of adventurous play as follows (number of ‘don’t know’ responses in brackets): Home = 47 (29); Outside at home = 66 (22); Playground = 80 (26); Green space = 154 (40); Near water = 428 (46); Indoor play centre = 164 (36); Street = 727 (37). Mean level of adventurous play across place was calculated with available items. For hours spent playing adventurously 131 participants (7%) had missing data due to a ‘don’t know’ response to at least one of the adventurous play rating questions.

### 2.5. Distribution and Outlier Checks

The distribution of all variables was checked prior to analyses. All of the total time spent playing in each place variables were Windsorized such that any values lower than the 5th percentile or higher than the 95th percentile were given the value of the 5th and 95th percentile respectively, which prevents outliers from substantially affecting results. All of the total time spent playing variables were very positively skewed. They were all, therefore, square-root transformed prior to analysis. The total time spent playing variables included outliers identified via boxplots with whiskers extending to 1.5 times the interquartile range. These variables were therefore also Windsorized. This process yielded data that were approximately normally distributed for all of the CPS variables. The Engagement with Risk scale of the REPS had some extreme low values and outliers were identified for the age allowed out alone question, both for child age and respondent age. These variables were, therefore, also Windsorized as described above.

## 3. Results

The results are presented under the heading of each research question. All analyses were conducted using the ‘survey’ package in R [27] which is designed for the analysis of weighted survey data. For all generalised linear models, the svyglm command was used. For research questions 4 and 5 sociodemographic factors and geographic factors are described as predictors, with children’s play the outcome variable, to be consistent with regression terminology. Given that the data are cross-sectional and observational, a significant result should not be taken as evidence that these factors are causally related to children’s play. Instead, a significant result informs us that there is an association between a specific factor and children’s play. The analyses therefor provide insight into which children are playing the most and which are playing the least.

### 3.1. Research Question 1: Where Do Children Aged 5–11 Years Living in Britain Spend Time Playing? Does the Amount of Time That They Spend Playing in Each Place Vary Significantly?

To address research question 1, three variables were used from the CPS: total time spent playing across the year, total time spent playing outside across the year, and total amount of time spent playing adventurously across the year. As expected, given there is an overlap in the items used to create the scores, these three measures from the CPS were all significantly correlated (*r*s ≥ 0.70, *p* < 0.001).

Children were reported to spend an average of 1140 h (*SD* = 641 h) playing per year. Of that time, 604 h (*SD* = 403 h), or 53%, was spent playing outside, and 133 h (*SD* = 133 h), or 12% of their total play time, was spent playing in nature.

Figure 1 shows the mean number of hours that children were reported to spend playing at each of the provided locations, across a year. The average total time children spent playing varied significantly across place, *F* (6,1912) = 958.37, *p* < 0.001. Coefficients demonstrated significant differences (at Bonferroni corrected alpha value of 0.002) between all places. Children spent the most time playing at home or at other people’s homes and the least time playing near water and at indoor play facilities, including swimming pools, trampoline parks and soft play. Away from home, children on average spent more time playing at playgrounds than in any other place.

### 3.2. Research Question 2: How Adventurously Do Children Living in Britain Play? Does This Vary by Play Location?

Figure 2 shows the mean level of adventurous play children engaged in across each play location. Children’s level of adventurous play varied significantly across place, F (6, 1889) = 218.07, *p* < 0.001. Coefficients indicated that the adventure level of each place differed significantly (at Bonferroni corrected alpha value of 0.002) from the adventure level of each other place with the exception of green space and indoor play centres, which were comparable. The highest levels of adventurous play happened in green spaces and at indoor play centres, which included soft play, trampoline parks and swimming pools. It is noteworthy that children’s play was reported to be most adventurous away from home and in the area close to home.

### 3.3. Research Question 3: What Age Are Children Living in Britain Allowed Out in Their Neighbourhood Alone?

A total of 108 respondents reported that they would not allow their child out alone and therefore did not state an age. Parents who did provide an age reported that their child was, or would be, allowed out alone at an average age of 10.74 years (SD = 2.20 years). This compares to parents’ report of the age when they were allowed out alone, which was at an average of age 8.91 years (SD = 2.31 years). A generalised linear model indicated that this difference was statistically significant, Beta = −1.83, SE = 0.05, t = −33.77, *p* < 0.001. Given that almost 6% of parents responded that they would not let their child out alone, this is likely to underestimate the actual mean age that children would be allowed out. For example, if we assume that these 108 respondents would allow their child out at age 14, that would increase the mean to 10.93 years, and if we assume age 16, that increases the mean to 11.05 years.

### 3.4. Research Question 4: To What Extent Are Socio-Demographic Factors, Geographic Factors and Parent Attitudes to Risk and Protection Associated with Children’s Total Time Spent Playing, Time Spent Playing Outdoors and Time Spent Playing Adventurously?

#### 3.4.1. Socio-Demographic Factors

The following socio-demographic variables were examined as predictors: child sex, child age, child birth-order, child disability, respondent health problem/disability in previous 12-months, respondent ethnicity, respondent employment status, respondent social class, respondent age, respondent education level. With the exception of child’s age, all of these variables were categorical. Some had low numbers of respondent within certain subcategories. To handle this and to simplify the analyses we collapsed across subcategories for a number of these variables as follows: ethnicity was collapsed into White/Non-white; birth order was collapsed into first born/not first born; education level was collapsed into low/medium/high, using the categorisation system used by YouGov; employment was collapsed into three categories employed full-time/employed part-time/unemployed and other (this final category included students, retired, unemployed, not working and other); parental age was categorised as younger/middle/older. To examine how these socio-demographic factors were associated with children’s total hours spent playing, hours spent playing outdoors and hours spent playing adventurously, three generalised linear models were fitted to the data. The results are shown in Table 2.

For total hours spent playing, the results indicate that child age, child sex, social grade, ethnicity, full time employment status (relative to working part time and not working/other) and respondent age were significant predictors. Parent disability status was not a significant predictor after Bonferroni correction for multiple comparisons. The children who played the most were younger and male, and their responding parent/caregiver was of lower social class, white, did not work full time and was relatively young in comparison to other respondents.

For hours spent playing outdoors, child sex, child disability, respondent health problem/disability, respondent full time employment status (relative to working part time) and respondent age were significant predictors of children’s time spent playing outdoors. The children who played outdoors the most were males who did not have a disability and whose responding caregiver was relatively young and worked part-time. Perhaps surprisingly, children whose responding parent/caregiver had a health condition or disability that significantly limited them spending more time playing outdoors than those whose parents were healthy or only limited a little by health or disability.

For time spent playing adventurously, child sex, child age, child disability, respondent health problem/disability, respondent full time employment status (relative to working part time) and respondent age were all significant predictors. The children who spent the most amount of time playing adventurously were boys, younger children, children who did not have a disability themselves and children whose responding parent/caregiver was white and working part-time. As with outdoor play, having a responding parent/caregiver with a limiting disability or health condition was related to more time spent playing adventurously.

#### 3.4.2. Geographic Factors

We examined two geographic predictors of children’s play: whether children lived in an urban, rural or town/fringe area and the location where children lived (seven categories: the five regions of England shown in Table 1, Scotland, Wales). To examine how these geographic factors were associated with children’s total hours spent playing, hours spent playing outdoors and hours spent playing adventurously, three generalised linear models were fit to the data. The results are shown in Table 3.

A significant main effect was found for region. To reduce the number of comparisons, we used Scotland, which had the highest play hours, and the East of England, which had the lowest play hours, as the reference categories, although results are only presented in the tables for Scotland as the reference to reduce the size of tables. Bonferroni corrected alpha of 0.004 was applied to these comparisons. The coefficients in Table 3 show that, relative to children in Scotland, children in the East of England spent significantly less time playing. Relative to children in the East of England, children in the South of England, and Scotland spent more time playing. For time playing outdoors, children in London, the North of England, the Midlands, and the East of England spent less time playing outdoors than children in Scotland. Relative to children in the East of England, children in the South of England, and Scotland spent more time playing outdoors. Time spent playing adventurously did not differ significantly across regions. For all three play variables, there were no significant differences between children living in an urban area and children living either in town/fringe areas or rural areas. It is important to note that, although these differences across regions are statistically significant, the proportion of variance accounted for by geographical locations overall is consistently less than 2%, indicating that regional differences are very small.

#### 3.4.3. Parental Attitudes to Risk

The REPS scale provides two scores, an Engagement with Risk subscale (*M* = 30.82, *SD* = 4.76) and a Protection from Physical Injury subscale (*M* = 26.07, *SD* = 5.20). There were six items on each scale, so these mean scores tell us that on average, for Engagement with Risk, respondents were more positive than negative. For Protection from Physical Injury, the average is close to the neutral response of the scale. Combined with the standard deviation this suggests that respondents varied in whether they felt children should or should not be protected from physical injury. These summary statistics are closely comparable to parent responses on the same measure reported in New Zealand and Canada [14,28].

Figure 3 shows the proportion of respondents belonging to each risk category, as categorised on the basis of the total score on the TRiPS. This shows that the majority of parents were relatively risk averse, with the most frequent category being somewhat risk averse.

We conducted three generalised linear models to examine whether total TRiPS score, as a measure of risk tolerance, and scores on the REPS subscales ‘Protection from Physical Injury’ and ‘Engagement with Risk’, predicted children’s total time spent playing, time spent playing outdoors and time spent playing adventurously respectively. The results are shown in Table 4. Total TRiPS score and the Engagement with Risk subscale score were both positively associated with the amount of time children spent playing; children whose parents reported they were more tolerant of risk and had more positive attitudes about their child engaging with risk were reported to spend more time playing. This was consistent across the total time spent playing, time spent playing outdoors and time spent playing adventurously. In contrast, scores on the Protection from Physical Injury scale of the REPS were not associated with any of the time spent playing variables.

### 3.5. Research Question 5: To What Extent Are Socio-Demographic Factors, Geographic Factors and Parental Attitudes to Risk and Protection Associated with Children’s Independent Mobility?

#### 3.5.1. Socio-Demographic Factors

To examine factors that predicted the age at which children were allowed out alone, three glm models were evaluated, one focusing on socio-demographic factors, one focusing on geographical factors, and a third focusing on parent/caregiver attitudes. The predictor variables used for all three analyses aligned with those used in the analyses previously presented, and the dependent variable was the age that respondent’s reported children were or would be allowed out alone. For sociodemographic factors (see Table 5), respondent ethnicity, respondent education level and birth-order predicted the age at which children were allowed out alone. Children who were white, not first born and whose parents had a higher level of education were allowed out at a younger age.

#### 3.5.2. Geographic Factors

For geographic factors (see Table 6), there were significant differences across regions; relative to children in Scotland, who were allowed out at the youngest age, children living in every region of England and Wales were significantly older when they were allowed out alone. Relative to children in the East of England, who were eldest when allowed out, children in London, the South of England, Wales and Scotland were younger when they were allowed out alone. The results indicate that children in Scotland were allowed out on average more than a year before those in Wales and all regions of England. In addition, relative to children living in urban environments, children living in towns and the fringes of urban areas were allowed out at a younger age. There was no significant difference between rural and urban areas.

#### 3.5.3. Parental Attitudes to Risk

Parent/caregiver attitudes to risk were also predictive of the age at which children were allowed out alone (see Table 7). Respondents who had lower scores on the Protection from Physical Injury scale and higher Tolerance of Risk scores had children who were allowed out alone at a younger age.

## 4. Discussion

In this paper we addressed five research questions focused on children’s play and independent mobility in Great Britain. This is the first paper to provide comprehensive, nationally representative data on children’s play in Britain. The findings provide important insights into where children play, how adventurously they play, and factors associated with play and independent mobility.

### 4.1. Research Question 1: Where Do Children Aged 5–11 Years Living in Britain Spend Time Playing? Does the Amount of Time That They Spend Playing in Each Place Vary Significantly?

The first research question focused on where children spend time playing. Our results showed that children spent on average 1140 h a year playing, which equates to an average of 3.12 h per day, although there is considerable variation between children. Consistent with previous research [9] and unsurprisingly, the place where children played the most was indoors at home or in other people’s homes. Outdoor play accounted for around half of children’s play and most commonly happened in gardens at home or in other people’s gardens. This is also consistent with previous research from Norway, showing that gardens are the most common outdoor space used for play [12]. Away from home, playgrounds were the most common place for children to play, followed by green spaces such as forests and grassy spaces and then on the street and local public spaces. This highlights the importance of public play spaces, such as playgrounds and green spaces, especially for those children who do not have access to a garden at home.

### 4.2. Research Question 2: How Adventurously Do Children Living in Britain Play? Does This Vary by Play Location?

Adventurous play has been described as beneficial for children’s fears and anxiety [6,7] so we addressed a second research question specifically concerning children’s adventurous play in different places. The findings indicate that children’s level of adventurous play differs across play spaces, but that this difference is relatively small, with all outdoor locations offering at least a mild level of adventure. Adventurous play was most likely to happen in green spaces, defined as trees, forests, woodland and/or grassy spaces, as well as indoor play centres, which included soft play, swimming pool and trampoline parks. This was closely followed by playgrounds and play near water. It is not surprising that the green spaces offered the highest level of adventurous play, as natural spaces by their very definition are not crafted with children’s safety in mind and they offer myriad opportunities for climbing, running, jumping and hiding. It is perhaps reassuring that children can access adventurous play and the accompanying feelings of thrill, excitement and fear, even if they do not have easy access to nature, at indoor play centres and particularly at public playgrounds, which are generally free and widely available. Concerns have been raised about whether public playgrounds offer children an appropriate level of challenge and there is evidence that some do not [29,30]. It is therefore vital that children’s play spaces are evaluated for the play opportunities, or affordances, that they offer, and not simply on the basis of maximising safety and minimising cost.

### 4.3. Research Question 3: What Age Are Children Living in Britain Allowed Out in Their Neighbourhood Alone?

Our third research question focused on the age that children were allowed out alone. Consistent with previous findings [10], we found a significant difference between the age that parents said they would allow their children out and the age at which they reported they were allowed out. The difference in means was almost two years, although there was substantial variation around the average age given and there is reason to think this may underestimate the age difference. This finding is of course limited by the fact that respondents are providing retrospective report of the age they had independent mobility and it is hard to estimate how accurate this is. The data do provide us with a baseline for future evaluations which will allow us to track over time whether the age that children are allowed out independently is changing. It is interesting to note the difference between our findings and those from the New Zealand State of Play Survey [14] which found that children were most often not allowed out without supervision until age 13 and not allowed out alone until age 15. Although our results show that the age children are allowed to be out alone in Britain has increased over the past generation, our results suggest that on average children were allowed out alone around their 11th birthday, which is substantially younger than for children in New Zealand.

### 4.4. Research Question 4: To What Extent Are, Socio-Demographic Factors, Geographic Factors and Parental Attitudes to Risk and Protection Associated with Children’s Total Time Spent Playing, Time Spent Playing Outdoors and Time Spent Playing Adventurously?

The final two research questions focused on how geographic location, socio-demographic factors and parent/caregiver attitudes were related to children’s play and independent mobility. An important starting point for discussing these findings is to highlight that none of these predictors accounted for a large amount of variance in children’s play or independent mobility. Geographic factors explained very little variance in children’s play (<2%) but were more important for independent mobility, explaining 5% of variance. In contrast, socio-demographic factors were the strongest predictor of children’s play, accounting for around 5–7% of variance but explained less than 1% of variance in independent mobility. Parent attitudes were the strongest predictor for independent mobility, accounting for around 9% of variance in the age that children were allowed out alone. They accounted for between 3–4% of variance in play measures, being a stronger predictor of adventurous and outdoor play than total play. This is perhaps not surprising given that the measures focused on risk tolerance which we would expect to be linked to children’s risk taking during adventurous play.

Taking geographical location first, as described this predicted very little variance in children’s play. Small regional differences were found, with children in Scotland playing slightly more, particularly outside, than children in other areas of Britain and children in the East of England playing the least. Interestingly this did not lead to more adventurous play in Scotland. No systematic differences were found between children living in an urban environment, a rural environment or in a town/fringe area. Previous research has shown that features of the built environment such as traffic, increased neighbourhood greenness and access to a yard are associated with children’s outdoor play [17]. Taken together with our findings, this suggests that it is not necessarily the type of environment a child lives in but specific features of that environment on a more local level. For example, one child might live in an urban environment but have access to a garden and low traffic areas whereas another might live in a rural environment where there is no access to a garden and where road safety is an issue.

Location was a more important predictor of independent mobility, and again it was children in Scotland who stood out, with Scottish children being allowed to be out independently at a younger age relative to all other included regions. Parents of children in the East of England reported that their child was/would be allowed out at the eldest age compared to all other regions; the average was almost two years older than the average age for children in Scotland. Children living in towns or on the fringes of urban centres were allowed out at a younger age than those living in urban and rural areas. This is consistent with previous research showing that children who live in towns make more journeys alone than children who live in rural villages or cities [31]. Although it was not a specific focus within this paper, it seems likely that this is due to availability of infrastructure on a local level that is perceived as improving children’s safety, such as pavements and streetlights and traffic volume [32,33], again highlighting the importance of children’s local environment.

For socio-demographic factors, a range of these were associated with children’s play and these differed by the type of play. Children played less and played less adventurously as they got older. Across all play variables, girls played less than boys, but this difference was largest for time spent playing adventurously. Children whose participating parent was from a lower social grade spent more time playing overall, but this effect was not found for outdoor or adventurous play, indicating that these children spend more time playing, but primarily at home or in other people’s homes. In contrast, child disability was only related to hours spent playing outside and adventurously; children reported to have a diagnosed learning difficulty, a mental health problem or a physical disability spent less time playing outdoors. Perhaps surprisingly, children whose responding parent/caregiver reported that they had a health problem or disability within the past 12 months played more across all measures than children whose responding parent did not have a health problem or disability. In general, children whose responding parent/caregiver was white played more than children with a non-white parent/caregiver, but only when looking across all play locations and not for outdoor or adventurous play, and children played more if their parent/caregiver worked part-time relative to full time and if their parent/caregiver was relatively young.

To our knowledge, only one study has previously examined predictors of children’s time spent outdoors in Britain [20]. In this study, correlations of time outdoors, rather than play specifically, were examined. Boys from a lower SES background who spent less than 2 h a day on a computer were found to spend more time outside. Our findings are only partially consistent with these; we found that children from lower SES backgrounds played more but SES was not a significant predictor of outdoor play. This inconsistency may be explained by our focus on play rather than time outdoors, our use of a nationally representative sample rather than a geographically limited opportunity sample, or the inclusion of other correlates of play within the same model that may explain some of the variance that might have been accounted for by SES. Our results are broadly consistent with previous international research. For example, Parent et al. [18] also found that play was associated with ethnicity and a recent review highlighted consistent associations between outdoor play and maternal employment status [19].

Socio-demographic factors were less relevant for independent mobility, with only respondent ethnicity, birth-order and respondent education level significant predictors; children whose responding caregiver was not white were older when they were allowed out, whereas children who were not first born and whose responding caregiver had completed a high level of education were younger when they were allowed out. Although the measures used and predictors examined vary, there are some similarities between our findings and previous research from the UK. For example, data from the Millennium Cohort Study also found that children had greater independent mobility if they were white [3]. This study also found that poverty was a positive predictor of independent mobility. In our results, children of respondents in the C2DE (lower SES) category were allowed out a younger age on average, but this difference was not significant. Instead, we found that higher levels of parent/caregiver education were predictive of earlier independent mobility, which appears to contrast with the findings of Aggio and colleagues. There are, however, substantial differences between the present study and the Millennium Cohort Study with respect to the way independent mobility is assessed; our measure of independent mobility is simply the age that children are allowed out rather than any estimate of how far they are allowed to travel and/or how frequently they do so. Further, the demographic predictors examined may explain differences between these study findings, with Aggio and colleagues [3] focusing only on child demographic factors, rather than parent/caregiver demographic factors, which we included here.

### 4.5. Research Question 5: To What Extent Are Socio-Demographic Factors, Geographic Factors and Parent Attitudes to Risk and Protection Associated with Children’s Independent Mobility

Parent/caregiver attitudes and beliefs about risk during play showed a small association with children’s play and a stronger association with independent mobility, although different factors were important. For play, parent/caregiver positive beliefs about risk, as measured by the engagement with risk subscale, and parent/caregiver tolerance of risk, were positively associated with the number of hours children spent playing, across all play measures. The effect was strongest for adventurous play. The protection from injury subscale was not associated with children’s time spent playing. In contrast, protection from injury was positively associated with the age that children were allowed out alone, meaning that parents who had stronger beliefs about protecting their child from injury let their children out alone at an older age. Respondents who had higher risk tolerance let their children out at a younger age. To our knowledge, this is the first study to directly examine associations between parent/caregiver attitudes to risk and children’s play and independent mobility, but the findings are in keeping with previous work showing that independent mobility was predicted by parent perceptions of safety and environment [21]. The results provide clear evidence that parent/caregiver attitudes and beliefs around children’s risk-taking are relevant in this context, particularly in relation to adventurous play and independent mobility.

Taken together the results provide an overall picture of children’s play in 2020, before the COVID-19 pandemic. Children, on average, were playing regularly, although there is huge variation between children. This variation is explained to some degree by socio-demographic differences, geographical factors and parent attitudes and beliefs but a substantial proportion of the variation between children was not accounted for. It seems likely that this is due to factors that were not measured in the survey such as child temperament, the safety and availability of play spaces locally to the child’s home and parental attitudes and beliefs about play more broadly. A significant strength of the study is that it provides a baseline which will allow future research to examine change in children’s play, use of different play spaces and independent mobility over time.

### 4.6. Limitations

Limitations of the study include the reliance on parent/caregiver report questionnaires. Although the test-retest and cross-informant reliability of the CPS has been examined [23], the measure will only ever provide an approximation of how much time children spend playing. Diary measures combined with activity trackers and GPS monitoring would give rich data on children’s play activities, which would complement the questionnaire measure used here. However, this type of data collection is costly, places a heavy burden on participants and it is only feasible to use for a short period of time. A further consideration regarding the measure is that the multiplication of responses to the time questions by responses to the frequency questions to get total time spent playing may lead to an overestimation; if parents slightly overestimate in their responses to both questions, these overestimates would be exaggerated through multiplication. There was also a substantial amount of missing data on a number of the demographic questions, due to participants opting not to respond to them. It is important that this option is provided for ethical reasons and the sample size, even with participants with missing data removed, was still adequate for the analyses conducted. A final limitation is that we did not break down the amount of time spent playing supervised or unsupervised within the CPS questions. The presence of adults affects children’s risk taking [34], aggression [35], physical activity [36] and social play [37]. It may therefore be of interest in future research to examine the proportion of play that is unsupervised. It will also be important for future research to begin to examine causal predictors of children’s play via longitudinal or experimental research; the results we present show evidence of association but should not be interpreted as indicating causal relationships. A significant strength of the study is the recruitment of a nationally representative sample, which was weighted back to the population, with these weights taken into consideration in the analyses. Further strengths include the range of data collected which allows a rich insight into children’s play in Britain.

## 5. Conclusions

The results of the British Children’s Play Survey presented here show that on average, children living in Britain in 2020 play for just over 3 h per day. Around half of children’s play happens outdoors. Away from home, playgrounds and green spaces are the most common places for children to play. The most adventurous places for play were green spaces, indoor play centres, including soft play, trampoline parks and swimming pools, followed by playgrounds and near water. A significant difference was found between the age that children are now allowed out alone in comparison to the previous generation, with children now almost two years older than their parents/caregivers were when granted independent mobility. A range of socio-demographic factors predicted children’s play, with the most consistent findings found for child age, child sex, parent age and parent employment status, with younger children whose responding parent was younger and worked part-time, playing the most. There was little evidence that geographic location had a substantial impact on children’s play, but it was important for independent mobility, with children living in town/fringe areas and children living in Scotland allowed out alone at a younger age. When parents/caregivers had more positive attitudes around children’s risk-taking in play, children spent more time playing and were able to be out of the house independently at a younger age.

## Figures and Tables

**Figure 1 ijerph-18-04334-f001:**
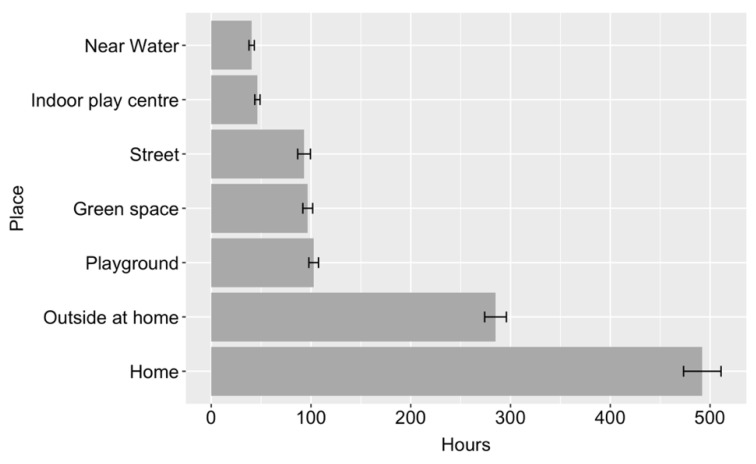
Mean time spent playing per year at each place (error bars show two standard errors).

**Figure 2 ijerph-18-04334-f002:**
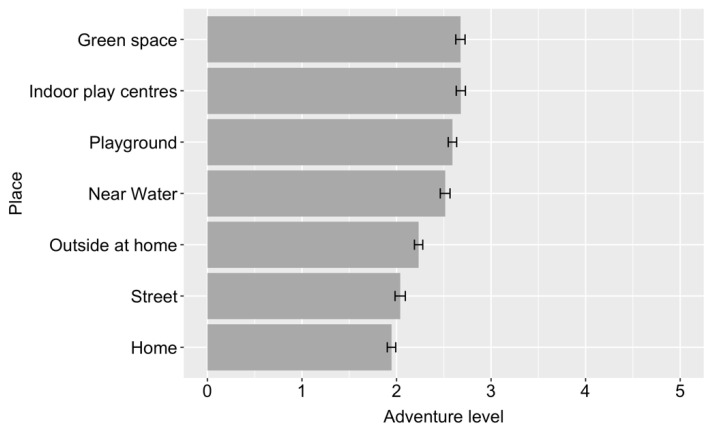
Mean level of adventurous play across place (error bars show two standard errors).

**Figure 3 ijerph-18-04334-f003:**
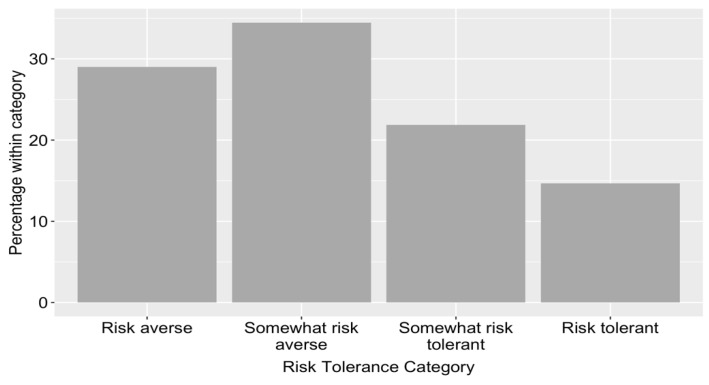
Proportion of parents belonging to each risk tolerance category as classified by the TRiPS.

**Table 1 ijerph-18-04334-t001:** Demographic characteristics of full sample.

Characteristic	*N* (%)
Parent sex	1919
Male	881 (46%)
Female	1038 (54%)
Child sex	1919
Male	982 (51%)
Female	937 (49%)
Parent age	1919
18–24	29 (2%)
25–34	370 (19%)
25–44	1026 (53%)
45–54	427 (22%)
54+	67 (3%)
Relationship to child	1919
Mother	990 (52%)
Father	805 (42%)
Stepparent	74 (4%)
Grandparent	32 (2%)
Others	18 (1%)
Child birth-order	1917 ^1^
First-born	1227 (64%)
Second-born	441 (23%)
Third or more	249 (13%)
Parent ethnicity	1556 ^1^
White British	1334 (86%)
White (other background)	81 (5%)
Black	24 (2%)
Asian	63 (4%)
Multi-ethnic	41(3%)
Other	13 (1%)
Employment status	1873 ^1^
Working full-time	1014 (53%)
Working part-time	418 (24%)
Student	29 (2%)
Retired	33 (2%)
Unemployed or not working	295 (15%)
Other	84 (4%)
Parent education level	1860 ^1^
Low	492
Medium	723
High	645
Marital status	1912 ^1^
Married, living as married, civil partnership	1527 (80%)
Separated, divorced or widowed	148 (8%)
Never married	237 (12%)
Full custody	1919
Yes	1828 (95%)
No	91 (5%)
GB Region	1919
England	1659 (85%)
North	470 (24%)
Midlands	338 (18%)
East	203 (11%)
London	203 (11%)
South	445 (23%)
Wales	86 (4%)
Scotland	174 (9%)
Location	1919
Urban	1521 (79%)
Town or Fringe	206 (11%)
Rural	192 (10%)
Parent social class ^2^	1919
Middle Class (ABC1)	1135 (59%)
Working Class (C2DE)	784 (41%)
Parent health problem/disability (within previous 12 months)	1592 ^1^
Yes, limits a lot	145 (9%)
Yes, limits a little	210 (13%)
No	1237 (78%)
Child disability ^3^	1919
Yes	243 (13%)
No	1597 (83%)
Prefer not to say	49 (3%)
Don’t know	30 (2%)

^1^ Some data are missing on this variable as participants chose not to provide this information. ^2^ The Market Research Society uses a demographic classification of social grade, which classifies families on the basis of the occupation of the head of the household. Social grade is typically used as a binary variable that categorises families as being either middle class or working class. This categorisation should be considered with relevant caveats in mind [22]. ^3^ defined as a ‘diagnosed learning disability, mental health disorder or physical disability’.

**Table 2 ijerph-18-04334-t002:** Socio-demographic predictors of hours spent playing, hours spent playing outside and hours spent playing adventurously per year.

	Hours Playing *p*/yr			Hours Playing Outside *p*/yr			Hours Playing Adventurously *p*/yr		
Predictors	Estimates	CI	*p*	Estimates	CI	*p*	Estimates	CI	*p*
(Intercept)	32.75	30.91–34.59	**<0.001**	23.90	22.26–25.54	**<0.001**	27.56	25.06–30.07	**<0.001**
Child age	−0.58	−0.86–−0.30	**<0.001**	−0.16	−0.41–0.08	0.192	−0.74	−1.13–−0.35	**<0.001**
Child sex: Male	Reference			Reference			Reference		
Child sex: Female	−1.21	−2.24–−0.17	**0.022**	−1.06	−1.97–−0.15	**0.022**	−2.23	−3.69–−0.78	**0.003**
Parent social grade: ABC1	Reference			Reference			Reference		
Parent social grade C2DE	1.28	0.15–2.42	**0.026**	0.47	−0.55–1.49	0.367	0.17	−1.42–1.76	0.834
Child disability: No	Reference			Reference			Reference		
Child disability: Yes	−1.16	−2.77–0.44	0.155	−1.61	−3.02–−0.20	**0.025**	−3.55	−5.91–−1.19	**0.003**
Parent health/disability: No	Reference			Reference			Reference		
Parent health/disability: Yes, limited a lot	2.15	0.23–4.06	**0.028**	3.72	1.93–5.51	**<** **0.001**	4.15	1.54–6.76	**0.002**
Parent health/disability: Yes, limited a little	0.84	−0.75–2.42	0.301	1.15	−0.30–2.59	0.120	1.28	−0.94–3.50	0.257
Parent ethnicity: White	Reference			Reference			Reference		
Parent ethnicity: Minority	−2.12	−3.57–−0.67	**0.004**	−0.58	−1.89–0.72	0.379	−1.32	−3.41–0.77	0.216
Employment: Full time	Reference			Reference			Reference		
Employment: Part time	2.27	1.01–3.53	**<** **0.001**	2.01	0.89–3.12	**<0.001**	2.34	0.56–4.12	**0.010**
Employment: Unemployed/other	2.12	0.69–3.55	**0.004**	0.95	−0.32–2.23	0.143	1.78	−0.24–3.80	0.084
Birth order: First born	Reference			Reference			Reference		
Birth order: Not first born	0.06	−1.10–1.22	0.918	0.44	−0.58–1.45	0.401	−0.90	−2.52–0.73	0.280
Parent age: Younger	Reference			Reference			Reference		
Parent age: Middle	−1.95	−3.42–−0.49	**0.009**	−1.49	−2.79–−0.19	**0.025**	−1.78	−3.84–0.27	0.089
Parent age: Older	−4.13	−5.76–−2.49	**<0.001**	−3.21	−4.70–−1.72	**<0.001**	−3.74	−6.03–−1.46	**0.001**
Parent education: Low	Reference			Reference			Reference		
Parent education: Med	−0.15	−1.53–1.22	0.826	−0.65	−1.87–0.56	0.294	0.38	−1.52–2.28	0.692
Parent education: High	0.95	−0.47–2.36	0.192	−0.69	−1.95–0.58	0.286	0.65	−1.37–2.68	0.526
Observations	1346			1346			1263		
R^2^/R^2^ adjusted	0.081/0.072			0.059/0.049			0.063/0.053		

*Note.* Bold *p*-values are <0.05, indicating statistical significance.

**Table 3 ijerph-18-04334-t003:** Geographic predictors of hours spent playing, hours spent playing outside and hours spent playing adventurously per year.

	Hours Playing *p*/yr			Hours Playing Outside *p*/yr			Hours Playing Adventurously *p*/yr		
Predictors	Estimates	CI	*p*	Estimates	CI	*p*	Estimates	CI	*p*
(Intercept)	33.78	32.23–35.32	**<0.001**	25.35	23.97–26.72	**<0.001**	27.43	25.15–29.72	**<0.001**
Region: Scotland	Reference			Reference			Reference		
Region: London	−1.43	−3.19–0.33	0.112	−2.47	−4.03–−0.90	**0.002**	−1.24	−3.82–1.34	0.348
Region: North	−1.70	−3.52–0.12	0.067	−3.07	−4.67–−1.48	**<0.001**	−2.08	−4.79–0.62	0.131
Region: Midlands	−1.67	−3.69–0.35	0.106	−2.81	−4.56–−1.06	**0.002**	−2.34	−5.28–0.59	0.118
Region: East	−3.69	−5.74–−1.64	**<0.001**	−4.38	−6.23–−2.53	**<0.001**	−2.31	−5.30–0.68	0.130
Region: South	−0.81	−2.57–0.94	0.365	−2.18	−3.72–−0.64	0.006	−0.48	−3.05–2.09	0.717
Region: Wales	−1.72	−4.49–1.05	0.224	−2.32	−4.74–0.10	0.060	−1.51	−5.29–2.26	0.432
Location: Urban	Reference			Reference			Reference		
Location: Town and fringe	−0.33	−1.82–1.16	0.663	0.71	−0.56–1.99	0.272	0.20	−1.91–2.31	0.853
Location: Rural	0.03	−1.45–1.51	0.969	0.81	−0.48–2.10	0.218	1.01	−0.86–2.88	0.291
Observations	1919			1919			1788		
R^2^/R^2^adjusted	0.008/0.004			0.016/0.012			0.004/−0.000		

*Note.* Bold *p*-values are <0.004, indicating Bonferroni-corrected statistical significance.

**Table 4 ijerph-18-04334-t004:** Parent attitude towards risk predictors of hours spent playing, hours spent playing outside and hours spent playing adventurously per year.

	Hours Playing *p*/yr			Hours Playing Outside *p*/yr			Hours Playing Adventurously *p*/yr		
Predictors	Estimates	CI	*p*	Estimates	CI	*p*	Estimates	CI	*p*
(Intercept)	32.27	31.83–32.72	**<0.001**	23.00	22.62–23.38	**<0.001**	26.20	25.58–26.83	**<0.001**
Engagement with risk	1.26	0.78–1.74	**<0.001**	0.81	0.38–1.24	**<0.001**	2.02	1.34–2.71	**<0.001**
Protection from injury	0.14	−0.35–0.64	0.577	−0.23	−0.65–0.19	0.274	−0.34	−1.07–0.38	0.355
TRiPs	0.87	0.33–1.40	**0.** **001**	1.21	0.76–1.67	**<0.001**	0.97	0.21–1.72	**0.** **013**
Observations	1919			1919			1788		
R^2^/R^2^ adjusted	0.030/0.029			0.042/0.040			0.037/0.035		

*Note.* Bold *p*-values are <0.05, indicating statistical significance.

**Table 5 ijerph-18-04334-t005:** Socio-demographic predictors of age children allowed out alone (independent mobility).

	Age Child Allowed Out Alone		
Predictors	Estimates	CI	*p*
(Intercept)	11.04	10.57–11.51	**<0.001**
Child sex: Male	Reference		
Child sex: Female	0.20	−0.04–0.44	0.099
Child age	0.03	−0.04–0.09	0.409
Parent social grade: ABC1	Reference		
Parent social grade: C2DE	−0.17	−0.44–0.10	0.214
Child disability: No	Reference		
Child disability: Yes	−0.06	−0.48–0.36	0.791
Parent health/disability: No	Reference		
Parent health/disability: Yes, limited a lot	−0.03	−0.54–0.48	0.909
Parent health/disability: Yes, limited a little	−0.15	−0.52–0.22	0.422
Parent ethnicity: White	Reference		
Parent ethnicity: Minority	0.44	0.04–0.84	**0.033**
Employment: Full time	Reference		
Employment: Part time	0.08	−0.21–0.38	0.578
Employment: Unemployed/other	0.16	−0.18–0.50	0.350
Birth Order: First born	Reference		
Birth Order: Not first born	−0.35	−0.61–−0.08	**0.010**
Parent age: Younger	Reference		
Parent age: Middle	−0.05	−0.41–0.32	0.803
Parent age: Older	0.00	−0.40–0.41	0.993
Parent education: Low	Reference		
Parent education: Med	−0.14	−0.46–0.19	0.406
Parent education: High	−0.47	−0.81–−0.13	**0.006**
Observations	1281		
R^2^/R^2^ adjusted	0.019/0.008		

Note. Bold *p*-values are <0.05, indicating statistical significance.

**Table 6 ijerph-18-04334-t006:** Geographic predictors of age children allowed out alone (independent mobility).

	Age Child Allowed Out Alone		
Predictors	Estimates	CI	*p*
(Intercept)	9.57	9.23–9.90	**<0.001**
Region: Scotland	Reference		
Region: London	1.09	0.70–1.47	**<0.001**
Region: North	1.55	1.14–1.95	**<0.001**
Region: Midlands	1.69	1.25–2.13	**<0.001**
Region: East	1.89	1.42–2.36	**<0.001**
Region: South	1.31	0.94–1.69	**<0.001**
Region: Wales	1.10	0.58–1.61	**<0.001**
Location: Urban	Reference		
Location: Town and fringe	−0.58	−0.88–−0.27	**<0.001**
Location: Rural	−0.22	−0.56–0.11	0.197
Observations	1811		
R^2^/R^2^ adjusted	0.058/0.054		

Note. Bold *p*-values are <0.05, indicating statistical significance.

**Table 7 ijerph-18-04334-t007:** Parental attitudes toward risk as predictors of age children allowed out alone (independent mobility).

	Age Child Allowed Out Alone		
Predictors	Estimates	CI	*p*
(Intercept)	10.76	10.66–10.86	**<0.001**
Engagement with risk	−0.10	−0.21–0.02	0.096
Protection from injury	0.16	0.04–0.28	**0.012**
TRiPS	−0.54	−0.67–−0.41	**<0.001**
Observations	1811		
R^2^/R^2^ adjusted	0.088/0.087		

Note. Bold *p*-values are <0.05, indicating statistical significance.

## Data Availability

The data used for the analyses presented in this study, along with the analysis script and full survey can be found here: http://doi.org/10.5255/UKDA-SN-8793-1.

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
