# Peer review of "Children’s Play and Independent Mobility in 2020: Results from the British Children’s Play Survey"

_ijerph, 2021, doi:10.3390/ijerph18084334_

Round 1
Reviewer 1 Report
I am pleased to see the study and, overall, I think it is a valuable contribution to our understanding of children's play in today's UK and the variable affecting that. I have a few methodological concerns that think would be worth considering for the authors. I will paste them in here:
Things to think about:
1. The survey questionnaire included a question about amount of time in outdoor play without supervision. This is not analyzed in the paper, perhaps because it would be zero in most cases, given that the children were 5-11 years old and the mean age that the parents said they would allow a child out alone was about 11 years. But it is an important question. This is the kind of play that is most missing in children’s lives in today’s world compared to the past. There is also research showing that when children play in the presence of an adult supervisor, the quality of their play is much reduced. It is less energetic, less creative, and, in a sense, less actual PLAY, as parents and other supervisors so frequently intervene. I think it would be useful at least for the authors to discuss this in the discussion section.
2. In calculating the average age of children at which parents reported they would allow their child out alone, the researchers discounted the 108 respondents who said they would not allow them out at any age. This results in an underestimation of the true average age. I did a rough calculation, using age 18 for those who said “never,” and found that including them in this way would result in an average of 11.14 years, rather than the 10.74 the researchers gave without including those. This makes the difference between the average age for their children and the average age that the parents were allowed out alone (8.91 years) considerably greater than reported in the paper. The difference is 2.23 years, not 1.83 years.
3. I have misgivings about the way you calculated total amount of play time. You did it essentially by multiplying the reported frequency for each type of play times the reported average amount of time per bout of that type and then adding this all together. Logically, that seems sound, but I suspect that it considerably inflates total play time. I think it is reasonable to suppose that respondents tend to slightly (at least slightly) overstate both frequency and time. They probably assume the researcher thinks play is a good thing, so, with imperfect memories and no record, they probably err a bit on the high side on these estimated. But multiplying two slight overestimates results in a large overestimate. Just for illustration, suppose the real frequency of a type of play is 4 times per week and the real average time devoted to each of these play bouts is 16 min. Now suppose the parent overestimates each of these by 25%--giving 5 times a week at 20 min. per time. Multiplying 4 X 16 results in 64 minutes; multiplying 5 X 20 results in 100 min. The result, instead of a 25% overestimate is a bit more than a 50% overestimate (100/64 = 1.56). I think it would have been much better to have simply asked parents first to estimate how much time (per week on average) their children played, total, and then, asked them to indicate how much of that play was of each type. You would still get overestimates, but they would not have been inflated by multiplication. You can't go back and re-do this, but it would be valuable to discuss the fact that your procedure may have resulted in large overestimates of total play time.
Author Response
I am pleased to see the study and, overall, I think it is a valuable contribution to our understanding of children's play in today's UK and the variable affecting that. I have a few methodological concerns that think would be worth considering for the authors. I will paste them in here:
Things to think about:
1. The survey questionnaire included a question about amount of time in outdoor play without supervision. This is not analyzed in the paper, perhaps because it would be zero in most cases, given that the children were 5-11 years old and the mean age that the parents said they would allow a child out alone was about 11 years. But it is an important question. This is the kind of play that is most missing in children’s lives in today’s world compared to the past. There is also research showing that when children play in the presence of an adult supervisor, the quality of their play is much reduced. It is less energetic, less creative, and, in a sense, less actual PLAY, as parents and other supervisors so frequently intervene. I think it would be useful at least for the authors to discuss this in the discussion section.
We agree with the reviewer that there is an important distinction between play that happens in the presence of a supervising adult and play that is independent. We included a separate question on this within the larger survey but chose not to include it in the paper for two reasons. First because the paper is already long and including the results from additional survey questions would lengthen it further. Second, because fewer than one-third of children were reported to play unsupervised more than once a week; the majority never played unsupervised, as the reviewer anticipated. Following the feedback from the reviewer we have now included some discussion of this within the limitations section of the discussion (see page 20, from line 696).
2. In calculating the average age of children at which parents reported they would allow their child out alone, the researchers discounted the 108 respondents who said they would not allow them out at any age. This results in an underestimation of the true average age. I did a rough calculation, using age 18 for those who said “never,” and found that including them in this way would result in an average of 11.14 years, rather than the 10.74 the researchers gave without including those. This makes the difference between the average age for their children and the average age that the parents were allowed out alone (8.91 years) considerably greater than reported in the paper. The difference is 2.23 years, not 1.83 years.
This is an important point. It is difficult to know how to interpret parents responding that they will never let their child out alone. It seems unlikely that parents would not allow their child out alone until age 18 so we estimated what the mean age would be if all parents who had responded with ‘never’ to this question actually let their child out alone at age 14 and at age 16 and we have added these estimates to the results on page 10 (see section 3.3). We have also mentioned in the discussion that there is reason to believe that the difference of almost 2 years between parent and child mean age could be an underestimation (p.17, from line 549).
3. I have misgivings about the way you calculated total amount of play time. You did it essentially by multiplying the reported frequency for each type of play times the reported average amount of time per bout of that type and then adding this all together. Logically, that seems sound, but I suspect that it considerably inflates total play time. I think it is reasonable to suppose that respondents tend to slightly (at least slightly) overstate both frequency and time. They probably assume the researcher thinks play is a good thing, so, with imperfect memories and no record, they probably err a bit on the high side on these estimated. But multiplying two slight overestimates results in a large overestimate. Just for illustration, suppose the real frequency of a type of play is 4 times per week and the real average time devoted to each of these play bouts is 16 min. Now suppose the parent overestimates each of these by 25%--giving 5 times a week at 20 min. per time. Multiplying 4 X 16 results in 64 minutes; multiplying 5 X 20 results in 100 min. The result, instead of a 25% overestimate is a bit more than a 50% overestimate (100/64 = 1.56). I think it would have been much better to have simply asked parents first to estimate how much time (per week on average) their children played, total, and then, asked them to indicate how much of that play was of each type. You would still get overestimates, but they would not have been inflated by multiplication. You can't go back and re-do this, but it would be valuable to discuss the fact that your procedure may have resulted in large overestimates of total play time.
Thank you for raising this issue, which we had not considered. We agree this could lead to overestimations. As the reviewer notes, we cannot go back and do this again. We have therefore included in the discussion some reflection on this point. See page 20 lines 688. When we were looking into this comment we realised that we had not removed outlying values from the estimates of time spent playing in each place, only the total hours spent playing variables used in the analyses. We have now removed outliers from estimates for each place and updated Figure 1 and the reported figures in the paper. This slightly reduces the mean estimates because the removed outliers were typically extreme high values.
Reviewer 2 Report
The manuscript is very extensive and contains a lot – maybe too much – information on outdoor play and independent mobility. Maybe I am wrong, but while reading I got the feeling that the manuscript is based on a study report or thesis. The authors present 5 research questions, of which number 4 and 5 may be of interest to readers outside of the UK. In the discussion of research question 4 and 5, I would expect a link to the introduction section. Instead a number of new references are presented. I do feel that leaving out research question 1 – 3 would increase readability of the manuscript.
INTRODUCTION
The introduction is very extensive. The authors present so much detailed information on outdoor play that the main goal of the study remains unclear. The authors could consider presenting the main findings for each heading in one paragraph. In order to shift the focus of the introduction more towards the study itself, the authors could add some more details of the British Children’s Play Survey (rationale, sample size).
Page 2 line 54 -61: the link between the information provided here and outdoor play is unclear.
METHODS
The statistical methods used in the study are not clearly described. The authors seem to have made predictive models, but if I understand the method section correctly the data were of a cross sectional cohort. I am no statistician, but this seems to be odd to me.
RESULTS and DISCUSSION
Both results and discussion sections are very extensive, the information provided to answer research question 1 exceeds the research question. While the discussion section is very extended, the limitation section is, in contrast, very short.
Author Response
The manuscript is very extensive and contains a lot – maybe too much – information on outdoor play and independent mobility. Maybe I am wrong, but while reading I got the feeling that the manuscript is based on a study report or thesis. The authors present 5 research questions, of which number 4 and 5 may be of interest to readers outside of the UK. In the discussion of research question 4 and 5, I would expect a link to the introduction section. Instead a number of new references are presented. I do feel that leaving out research question 1 – 3 would increase readability of the manuscript.
The manuscript is not based on a study report or thesis and is approximately the same length as comparable studies published within this journal (e.g. Jelleyman et al., 2019). We can see that questions 1 to 3 are of greater interest to readers in the UK than internationally but we believe they also provide important data for supporting international comparisons as well as future work tracking changes and trends in children’s play in western society. We intentionally presented the results under subheadings aligned with the research questions to ensure that readers could skip to the research questions they were most interested in. In response to the reviewer’s comment we have removed the additional analyses that we included for Q1 to reduce the length of the results and removed the corresponding discussion sections. We have also edited the discussion in relation to research questions 4 and 5 to ensure we refer back to the literature presented in the introduction. There are new references introduced at this stage given that some of the results were not anticipated prior to conducting the study and because we also wanted to ensure the introduction focused on the overall purpose of the research.
INTRODUCTION
The introduction is very extensive. The authors present so much detailed information on outdoor play that the main goal of the study remains unclear. The authors could consider presenting the main findings for each heading in one paragraph. In order to shift the focus of the introduction more towards the study itself, the authors could add some more details of the British Children’s Play Survey (rationale, sample size).
Page 2 line 54 -61: the link between the information provided here and outdoor play is unclear.
Following the reviewer’s comment we have reduced the length of the introduction by 30%. It is now 1281 words. Whilst making these edits we changed the first paragraph to introduce the British Children’s Play Survey earlier. We have not included more detail about the British Children’s Play Survey within the introduction because this would repeat information that belongs in the method and lengthen the paper unnecessarily. The section about mental health problems in children has now been removed.
METHODS
The statistical methods used in the study are not clearly described. The authors seem to have made predictive models, but if I understand the method section correctly the data were of a cross sectional cohort. I am no statistician, but this seems to be odd to me.
We are confident that our approach to analysis is entirely appropriate. The statistical models are predictive models but we do not imply that the relationships are causal; they are models of association. Prediction is simply used to describe relationships, and this can be done either between variables at the same time point or between variables. To give an example, we are not claiming that it is because someone lives in Scotland that their children are playing outdoors more (although this might be the case), but rather that if someone lives in Scotland, then it is more likely that their children are playing outdoors more. Further, it is relevant to consider that many of the predictor variables cannot logically be affected by children’s play (e.g. ethnicity, sex, age). We have now included an explanation regarding our analytical approach at the beginning of the results (see page 9, from line 323). Further, in the discussion we include as a limitation that our data do not provide data on causal relationships (see page 20, from line 700).
RESULTS and DISCUSSION
Both results and discussion sections are very extensive, the information provided to answer research question 1 exceeds the research question. While the discussion section is very extended, the limitation section is, in contrast, very short.
We have removed some of the additional analyses from the results so that they are focused only on the specified research questions. We have also removed the discussion about these additional findings, which makes it more focused. In addition, we have significantly expanded the limitations section. Despite this, the discussion is now shorter than in the original manuscript.
Round 2
Reviewer 2 Report
The authors did a great job in reducing the length of the manuscript. I am still of the opinion that the manuscript is very extensive, the argument that other manuscripts have a comparable length did not change my mind on that respect. But if the editor likes to give this manuscript a go then I’m fine with that.
The authors reply that they intentionally present subheadings for each of the research questions to ensure that readers could skip to the information of interest. I would suggest to also add those subheadings to the discussion section (if at all possible).
Author Response
We have added headings to the discussion to better guide the reader.